# DEAD-Box RNA Helicases and Genome Stability

**DOI:** 10.3390/genes12101471

**Published:** 2021-09-23

**Authors:** Michael Cargill, Rasika Venkataraman, Stanley Lee

**Affiliations:** 1Clinical Research Division, Fred Hutchinson Cancer Research Center, Seattle, WA 98109, USA; rasikav@uw.edu; 2Department of Laboratory Medicine and Pathology, University of Washington, Seattle, WA 98195, USA

**Keywords:** DEAD-box RNA helicases, genome stability, cancer

## Abstract

DEAD-box RNA helicases are important regulators of RNA metabolism and have been implicated in the development of cancer. Interestingly, these helicases constitute a major recurring family of RNA-binding proteins important for protecting the genome. Current studies have provided insight into the connection between genomic stability and several DEAD-box RNA helicase family proteins including DDX1, DDX3X, DDX5, DDX19, DDX21, DDX39B, and DDX41. For each helicase, we have reviewed evidence supporting their role in protecting the genome and their suggested mechanisms. Such helicases regulate the expression of factors promoting genomic stability, prevent DNA damage, and can participate directly in the response and repair of DNA damage. Finally, we summarized the pathological and therapeutic relationship between DEAD-box RNA helicases and cancer with respect to their novel role in genome stability.

## 1. Introduction

RNA helicases of the DEAD-box family belong to the helicase superfamily 2 (SF2), one of the largest families of RNA helicases present in eukaryotes, archaea and bacteria [1]. DEAD-box RNA (DDX) helicases are characterized by the presence of a specific amino acid motif, Asp-Glu-Ala-Asp (DEAD). DDX helicases comprise highly conserved core domains that regulate substrate binding, ATPase activity, and RNA unwinding helicase activity, though the catalytic activity may vary among DDX helicase [2,3]. These core domains are also highly conserved among family members, in contrast to the N- and C-terminal regions which are variable and divergent in length and composition (Figure 1A). The specific roles of these variable terminal regions are still largely unknown and are thought to confer functional specificity to these helicases.

DDX helicases are often essential for cell viability and have been implicated in disease [4]. DDX helicases play a central role in all aspects of cellular RNA metabolism from transcriptional regulation to translation initiation. Several cancers display altered expression levels and mutation of DDX helicase genes. Additionally, germline mutations have pleiotropic effects including neurodegeneration, dysregulated hematopoiesis, and a predisposition to cancer [5].

Recent studies have highlighted a link between RNA metabolism and genomic stability [6,7,8]. RNA-binding proteins appear to function both in the expression of genes important for genome stability and as active participants. DDX helicases belong to the growing number of RNA-binding proteins important for maintaining the genome. There are over 35 DDX helicase members in humans which are highly conserved in structure and arise from closely related nodes on reconstructing a phylogenetic tree with molecular sequences (Figure 1B) [9]. In this review, we will summarize the evidence supporting this novel function of DDX helicases, focusing on seven members whose roles in promoting genome integrity have been highlighted in recent studies. Specifically, these DDX helicases have both confirmed roles in RNA metabolism and a direct link to genomic instability in human cells. Finally, we provide insight into the association of aberrantly expressed DDX helicases in different cancers, implications of their roles in cancer genome maintenance, and development of therapies.

**Figure 1 genes-12-01471-f001:**
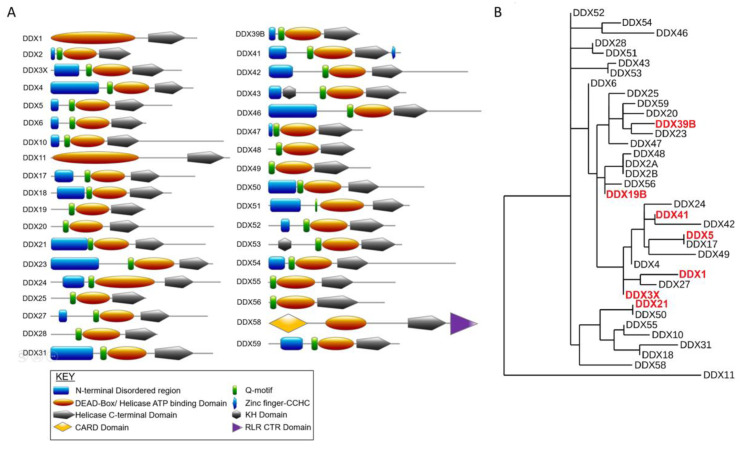
Human DEAD-box helicase family. (**A**) Protein domain schematic for each DDX helicase was generated with the PROSITE image creator tool MyDomains [10]. (**B**) The phylogeny tree for human DEAD-box helicase proteins was generated with Phylogeny.fr. Helicases described in this review with a role in promoting genome integrity are highlighted in red.

## 2. DEAD-Box RNA Helicase Studies

### 2.1. DDX1

DDX1 is one of the earliest human DDX helicases investigated to have an active role in genomic stability. γ irradiation of HeLa cells revealed extensive colocalization of DDX1 and γH2AX concurrent with an overall increase in DDX1 nuclear levels [11]. Colocalization of DDX1 at sites of irradiation-induced damage was ATM-mediated and was dependent on the presence of DNA:RNA hybrids. Mechanistically, DDX1 depletion caused radiosensitivity and an increase in DNA breaks in human cells [12]. Fluorescent reporter studies suggest that an impaired homology directed repair (HDR) underpins this phenotype. Specifically, the CtIP-mediated generation of resected ends is not dependent on DDX1, while the subsequent loading of downstream HDR factors, RPA, Rad51, and recently BLM is affected [13]. Using DNA:RNA hybrid immunoprecipitation (DRIP) qPCR, levels of DNA:RNA hybrids at resected ends induced by the endonuclease, I-SceI, are increased upon DDX1 depletion [12]. These lines of evidence support a model whereby DDX1 is recruited to DNA breaks to remove DNA:RNA hybrids that arise at resected DNA during HDR. It is thought these RNA species serve a regulatory function in DNA repair—a notion which is supported by yeast studies [14].

DDX1 studies have also introduced a novel role for DDX helicases in antigen receptor recombination. Interestingly, DNA:RNA hybrids were first described at murine immunoglobulin (Ig) loci [15]. However, instead of resolving DNA:RNA hybrids, DDX1 is able to promote DNA:RNA hybrid formation at the murine Ig locus [16]. Conditional Ddx1 knockout mice display aberrant class switch recombination (CSR) as evidenced by reduced IgG generation, and AID binding in the switch region is reduced without Ddx1 in a CSR cellular model. Interestingly, Ddx1 depletion reduces DNA:RNA hybrid levels across the constant elements in the Ig-heavy locus. In vitro analysis revealed that Ddx1 binds and unwinds G-quadruplexes (G4) present in intronic Ig RNA, which, in turn, are able to hybridize with complementary DNA. Such findings suggest a model in which DDX1 unwinds complex RNA to form DNA:RNA hybrids. It is tempting to speculate whether this mechanism exists more broadly as a part of DNA:RNA hybrid regulation important for maintaining the genome.

Finally, DDX1 interacts with Drosha in an RNA-dependent manner to generate microRNA [17]. Upon neocarzinostatin treatment, U2OS cells generate a number of microRNAs in a DDX1-dependent manner. Evidence suggests that DNA damage-induced miRNA are used to rapidly modulate the expression of DNA damage and response genes [18]. DDX1-dependent species include miR-138 and miR-101, which are known to inhibit DNA damage signaling proteins ATM, DNA-PKcs, and γH2AX [19]. While the function of DDX1 in miRNA biogenesis is unclear, it remains a route of investigation for its role in genomic stability. Overall, studies of DDX1 have served as a cornerstone in our understanding of the many roles that DDX helicases play in genomic stability.

### 2.2. DDX3X

*DDX3X* is an X-linked gene important for RNA export, translation, and ribosomal RNA (rRNA) biogenesis [20]. DDX3X and its homologs are essential for yeast, murine, and human cell viability [21,22]. Microscopic analysis of *Ddx3x*-deficient murine embryonic cells shows an increase in DNA damage and stimulated p53, pointing to a link between Ddx3x and genome stability. Disruption of DDX3X through small-molecule inhibition and gene knockdown in human cancer cell lines similarly leads to an accumulation of DNA damage [23]. The underlying mechanism of DDX3X in maintaining the genome remains an active area of investigation. While DDX3X is able to unwind DNA:RNA hybrids in vitro, it is unknown whether DDX3X similarly processes cellular DNA:RNA hybrids [24,25]. Fascinatingly, DDX3X displays RNaseH2 activity in vitro and is suggested to participate in the excision of genomic ribonucleotides [26].

DDX3X may also regulate DNA repair gene expression as *POLM* and *MRE11A* transcript levels are reduced upon DDX3X disruption [23]. Similarly, Ddx3x knockout in murine hepatocytes leads to reduced expression of DDB2 and XPA as confirmed at the protein level [27]. Finally, DDX3X promotes the processing of microRNA species via Drosha/DGCR8 [28]. Pathway analysis of microRNAs affected by DDX3X levels revealed an enrichment in cell cycle checkpoint genes, suggesting that DDX3X may regulate the cell cycle through microRNA interference.

Interestingly, DDX3X may play an active role in the response and repair of DNA damage. With respect to response, DDX3X is important for stabilizing p53 to promote DNA damage-induced apoptosis, thereby limiting chromosomal abnormalities [29,30]. Regarding direct DNA damage interaction, live-cell imaging revealed that DDX3X is recruited to sites of microirradiation in a PARP1-dependent manner [31]. This is a consistent model by which a growing number of RNA-binding proteins are localized to sites of DNA damage [32]. DDX3X studies reflect the multifaceted roles that DDX helicases may play in promoting genomic stability.

### 2.3. DDX5

DDX5 is a known transcriptional regulator [33]. DDX5 knockdown reduces the protein levels of a number of DNA replication proteins [34]. Coimmunoprecipitation analysis suggests that DDX5 is important for loading RNA polymerase to initiate the transcription of such DNA replication genes. This role in transcriptional regulation is thought to underpin a DNA replication-deficient phenotype as measured by episomal maintenance assays in cancer cells. Furthermore, DDX5 was found to immunoprecipitate with p53 and can function as a co-activator at p53-dependent promoters [35]. Upon DNA damage by etoposide treatment or γ irradiation, DDX5 mediates recruitment of p53 to the p21 gene important for the G1/S checkpoint [36]. Subsequently, Ddx5 knockout radiosensitizes p53-dependent murine bone marrow cells. DDX5 depletion causes radiosensitivity and etoposide sensitivity in human cells [37].

DDX5 can also resolve DNA:RNA hybrids [37,38]. In human cells, DDX5 depletion is associated with an increase in global DNA:RNA hybrid levels particularly in regions of active transcription [38,39]. DDX5 may resolve DNA:RNA hybrids as a transcriptional regulatory mechanism with implications for genomic stability. In fact, DDX5 depletion causes DNA damage and confers sensitivity to hydroxyurea, suggesting that DDX5 is important for protecting against replication stress.

Additional lines of investigation suggest that DDX5 is also an active participant in the repair of damaged DNA. DDX5 may be involved with DNA break repair as proximity-ligation mapping revealed an interaction with Ku, an essential component of non-homologous end joining repair [40]. In live-cell imaging, DDX5 is actively excluded from DNA damage in an ATM-dependent manner [37]. It is interesting to note that the RGG motifs in DDX5 are important for R-loop and DNA damage localization, reminiscent of liquid-liquid phase separation [37,38]. This is the prevailing mechanism by which RNA-binding proteins have been found to be recruited to sites of damage [32]. Inhibiting transcription and digesting DNA:RNA hybrids allows for the recruitment of DDX5 which was additionally found to bind transcripts upon break induction. Furthermore, site-specific qPCR detected more DNA:RNA hybrids upon break induction in DDX5-depleted cells, and an interaction with BRCA2 appears to mediate this pro-repair function of DDX5 [41]. Such findings imply DDX5 binds and removes DNA:RNA hybrids from sites of DNA damage. Finally, DDX5 depletion causes transcription-associated deletions at break sites as well as delayed Rad51 kinetics, pointing to an importance for homology-based break repair [37,41].

Of note, DDX5 shares significant homology with DDX17 which also functions as a p53 interactor and transcriptional regulator [33,35]. DDX17 was also found to interact with the canonical DNA break repair protein, Ku [40]. While less is known about this homologous DDX helicase, an siRNA screen in a human reporter line suggested that DDX17 can negatively regulate DNA break repair by HDR [42]. A recent study found that overexpression of DDX17 promoted the repair of etoposide-induced breaks in human cells [43]. Further studies on DDX5 and DDX17 may provide an evolutionary lens on the relationship between DDX helicases and genome stability.

### 2.4. DDX19

DDX19 is a well-studied DDX helicase with numerous cellular functions. Regarding genomic stability, depletion of DDX19 in human cells causes the accumulation of DNA breaks [44,45]. Studies suggest that DDX19 likely protects against R-loop-mediated replication stress. UV irradiation triggers the relocation of DDX19 into the nucleus in an ATR- but not ATM-dependent manner [44]. DDX19 depletion limits DNA replication, but can be rescued upon RNaseH expression. DNA:RNA hybrid levels are sensitive to DDX19 both in vitro and in vivo, highly suggesting that DDX19 can directly unwind these structures. Furthermore, DDX47, a highly similar helicase, may be recruited by FANCD2 to resolve cellular R-loops [45].

Gene expression analysis by real-time PCR and microarray suggests another role for DDX19 promoting genomic stability. In particular, DDX19 depletion reduces *BRCA1* and *FANCD2* RNA levels, both of which are associated with protecting against replication stress [46]. However, it is unknown how DDX19 regulates the expression of these genes and if their protein levels are ultimately altered.

### 2.5. DDX21

DDX21 is a predominantly nucleolar helicase whose genomic stability function is currently linked to R-loop-associated stress [47,48]. DDX21 depletion causes DNA damage in an RNaseH-dependent manner as seen by both microscopy and comet assay [49]. DDX21 may directly resolve R-loops as genes that immunoprecipitate with DDX21 accumulate DNA damage and DNA:RNA hybrids upon DDX21 depletion. Nucleolar DNA:RNA hybrid levels are also increased. Furthermore, communication between nucleolar DDX21 and the DDR has been established through microscopic studies in human cells [50]. DNA break induction of rDNA by I-Ppol causes the exclusion of DDX21 from the nucleolus and is dependent on DNA-PKcs and ATM activity. As numerous DDX helicases are important for ribosome biogenesis in nucleoli, DDX21 studies highlight the potential importance of DDX helicases on nucleoli stability [3].

### 2.6. DDX39B

The role of DDX39B in genomic stability was established through studies of Sub2—A yeast DExH RNA helicase homolog. Sub2/DDX39B are components of the THO complex important for the processing and export of mRNA. Using a yeast recombination reporter system, deletion of Sub2 induced a hyper-recombination phenotype as does ablation of other THO complex members such as mft1 and hrp1 [51,52,53]. Furthermore, hrp1 KO-mediated hyper-recombination was found to be transcription-dependent and can be rescued upon ectopic expression of Sub2 [52]. Further investigation of mRNA export mutants suggests that co-transcriptional R-loops underpin the hyper-recombination phenotype [54]. Indeed, expression of Sub2 catalytically-deficient mutants induce hyper-recombination in a dominant negative manner [55].

In humans, DDX39B (UAP56) depletion increases endogenous DNA damage [56,57,58]. The DNA damage phenotype was found to be dependent on the helicase activity [58]. Importantly, transcription inhibition by cordycepin and DNA:RNA hybrid degradation by RNaseH rescued the genomic instability phenotype in UAP56-depleted cells [57]. UAP56 depletion appears to generate DNA damage during DNA replication due to collisions between DNA replication machinery and unresolved R-loops. Conversely, overexpression of UAP56 can rescue R-loop-mediated DNA damage in canonical models such as Senataxin (DNA/RNA helicase)-depleted cells. Overall, yeast and human models studies collectively suggest that DDX39B can prevent genomic instability by resolving R-loops.

Interestingly, DDX39B depletion causes DNA damage at telomeres as visualized by in situ hybridization of telomeric repeats [58]. The depletion or overexpression of DDX39B reduces or extends telomeres, accordingly, and interacts with the telomere extending domain, TRF2. Telomeric maintenance intersects with R-loop metabolism via the generation of telomeric repeat-containing RNA (TERRA) [59]. This RNA species hybridizes with telomeric DNA to regulate telomere length. It is tempting to speculate that DDX39B is important for TERRA metabolism.

DDX39B may also prevent genomic instability independent of its role in processing DNA:RNA hybrids. In the absence of rad7-dependent global excision repair, Sub2 deficiency causes UV sensitivity [60]. The repair of UV damage DNA on the transcribed strand of the constitutive RPB2 gene is reduced in Sub2-deficient cells, together suggesting that Sub2 promotes transcription-coupled repair (TCR). In this model, however, ribozyme-mediated cleavage of nascently transcribed RNA does not rescue TCR deficiency, which likely precludes DNA:RNA hybrids from underpinning the phenotype. Instead, the data suggest that RNA polymerase stalling from disrupted mRNA export alone can block TCR machinery. Thus, Sub2 likely functions to evict RNA polymerase at sites of DNA damage.

Finally, DDX39B can promote genomic stability via DNA repair gene expression. DDX39B depletion causes sensitivity to a number of DNA damaging treatments including cisplatin, mitomycin C, and γ irradiation [61]. DDX39B regulates the expression of BRCA1 likely through mRNA stabilization and export. Subsequently, DDX39B ablation reduces DNA resection at DNA breaks, thereby reducing homology-directed repair.

### 2.7. DDX41

DDX41 has multiple cellular functions associated with innate immune signaling, inflammation, R-loop metabolism and splicing [62,63,64,65,66]. Disrupting DDX41 in human and zebrafish models is associated with the accumulation of DNA breaks [63,67]. In zebrafish, *ddx41* loss-of-function mutants (*ddx41^sa14887^*) develop anemia due to defective erythropoiesis. Mechanistically, this was attributed to increased DNA damage response in *ddx41* mutants which resulted in ATM and ATR-triggered cell cycle arrest of erythroid progenitors. RNA-seq and pathway analysis on *ddx41* mutant cells identified downregulation of genes primarily involved in mRNA splicing, cell cycle arrest, DNA repair and DNA replication, for example: cdkn1a, mdm2 and mcm10.

Interestingly, DDX41 was also identified in an R-loop interactome, suggesting a role in the regulation of DNA:RNA hybrids [65,67]. Detection of R-loops via immunofluorescence demonstrated that R-loops levels were increased in ddx41 loss-of-function mutants in zebrafish embryos, which could be reversed upon RNaseHI expression [63]. Similar to findings in zebrafish, DDX41 was found to suppress R-loop accumulation in human HEK-293 cells, suggesting a possible conservation of mechanism [67]. Resolving R-loops appears important for the genomic stability function of DDX41 as overexpression of RNaseH1 in these cells partially rescued the accumulation of DNA breaks upon DDX41 knockdown. Of significance, shRNA-mediated depletion of DDX41, or expression of DDX41 variants in DEAD (L237F/P238T) or helicase domain (R525H) found in AML patients in human CD34+ HSPCs led to an increase in R-loops and 53BP1 foci, suggesting that DDX41 mutation may contribute to human disease. Mechanistically, DDX41 wild-type protein, but not the R525H helicase mutant, was able to successfully bind and unwind DNA:RNA hybrids in vitro, further confirming its active role in regulating R-loops.

DDX41 may also play a role in telomere maintenance which is often associated with R-loop processing factors [68]. Peripheral blood cell telomere length measured by monochrome multiplex quantitative PCR was found to be shorter in a cohort of patients harboring germline DDX41 variants compared to age-matched controls. Of note, the *C. elegens* homolog, SACY-1, is important for lifespan [69]. As with DDX39B, it is possible that DDX41 is important for regulating TERRAs.

Finally, DDX41 also regulates the expression of p21 [70]. Despite its predominant role in splicing, DDX41 appears to affect the expression of p21 through an interaction with its 3′ UTR as a post-transcriptional negative regulator. Furthermore, knockout of p53 ablates this relationship, suggesting that DDX41 acts within the p53 signaling axis, though irrespective of DNA damage upon irradiation.

## 3. Implications for Cancer

The DDX family of RNA helicases are multifunctional proteins with ubiquitous functions in RNA metabolism such as pre-mRNA splicing, miRNA processing, RNA transport, and ribosome biogenesis [1]. Multiple reviews have discussed the role of DDX helicases in cancer [4,71]. However, to our knowledge, this is the first review highlighting the role of DDX helicases in promoting genome stability. DDX helicases play a crucial role in the maintenance of genome integrity through several mechanisms such as regulating gene expression, processing structures such as R-loops, and participating directly in the DNA damage repair response (Figure 2). Genomic instability is a potent driver of clonal evolution in cancer, thereby facilitating the acquisition of other hallmarks of cancer such as the deregulation of cellular metabolism and evasion of immune recognition [72]. Indeed, DDX23 appears important for processing DNA:RNA hybrids to prevent genome instability, a novel function which has directly been linked to the development of adenoid cystic carcinoma [73]. This novel role for DDX helicases may also provide anti-cancer therapeutic strategies. For example, DDX54 knockdown radiosensitizes breast cancer cells likely due to the role of DDX54 in regulating gene expression upon irradiation [74,75].

Over the past decade, high-throughput next-generation sequencing of cancer genomes has associated aberrant expression of DDX helicases with many cancers. Reviews summarizing large-dataset analyses from consortia such as The Cancer Genome Atlas (TCGA) have highlighted the various aberrations seen in DDX helicase genes across various cancers including breast, lung, leukemia, and melanoma [71]. For instance, increased tumor malignancy and poor treatment outcome in non-small-cell lung carcinoma were associated with loss of DDX3X [76]. DDX5 and DDX17 have been associated with multiple cancers such as breast, colon, and pancreatic cancer, where they transcriptionally regulate the activity of p53, estrogen receptor-α and β-catenin [77,78]. More recently, increased DDX21 expression was observed in breast cancer, where it was associated with increased c-Jun activity and rRNA processing, driving breast tumorigenesis [44]. Germline mutations of DDX41 promote the development of myelodysplastic syndromes (MDS) with age, and germline mutations predispose individuals to somatic secondary mutation of DDX41 in the healthy allele which is strongly associated with the development of secondary AML [66]. DDX39 was found to be upregulated in hepatocellular carcinoma tissues and cells, and its expression was positively correlated with cancer metastasis and poor survival outcome in patients with high-DDX39 expression [79]. A similar correlation in DDX39 expression levels and poor outcomes was observed in estrogen receptor positive breast cancer patients [80].

The discovery of aberrations in DDX helicases in the context of their role as gatekeepers of genome integrity highlights their relationship in cancer development. As discussed here, DDX helicases regulate metabolism of tertiary nucleic acid structures such as R-loops or G4. If left unregulated, such nucleic acid species may induce transcription and/or replication stress, mutagenesis, and may be a source of immunogenic cytosolic chromatin fragments [81,82,83]. A recent review highlighting the significance of genome integrity-related dysfunctions and molecular determinants of immunity underscores the importance of targeting DNA damage response and associated molecular players such as cGAS-STING activation to complement immune-checkpoint inhibition in cancer [84]. In relation to this, a recent study in the zebrafish model, identified that loss of *ddx41* led to an increase in cGAS-STING-mediated inflammatory signaling via the accumulation of R-loops [64]. Further investigation of these helicases via screens, high-throughput assays and multiomics approaches in the context of genome stability could identify potential new DDX helicases as biomarkers for diagnosis and prognosis, or as pharmacological drug targets for cancer. One such TCGA screen of candidate genes in colorectal cancer identified that loss of DDX56 reduced intron retention and tumor suppressor WEE1 expression, which functions as a G2-M DNA damage checkpoint, postulating that DDX56 could serve as a potential prognostic biomarker of colorectal cancer [85]. Furthermore, therapeutic strategies such as small molecules and inhibitors against the ATPase, helicase, and RNA-binding activity of DDX helicases have been explored for their anti-neoplastic abilities. RX-5902, an inhibitor that binds directly to phosphorylated DDX5, a post-translational modification of DDX5 which mediates epithelial to mesenchymal transition only in transformed cancer cells, is being studied in Phase I clinical trials for treatment of triple negative breast cancer [86]. More recently, RX-5902 was found to enhance the efficacy of immune checkpoint inhibitors in preclinical studies of triple negative breast cancer [87]. Targeting of DDX3 has also appeared to be efficacious. DDX3 overexpression is associated with poor survival in lung cancer, and the development and use of the DDX3 inhibitor, RK-33, showed tumor regression by inducing apoptosis of tumor cells in a xenograft model of lung cancer and was associated with an accumulation of DNA damage [23]. This vulnerability has been exploited in a breast cancer model where RK-33 inhibition showed therapeutic synergy with Olaparib in killing breast cancer cells [88]. Targeting of DDX helicases appears effective and may provide multiple vulnerabilities for combination therapies.

## 4. Concluding Remarks

Taken together, these studies provide compelling evidence to further identify and investigate other DDX helicase genes and associated alterations that may drive cancer through perturbations of the genome. For example, DDX42 and DDX50 were found to be PARP1 interactors, DDX26B is a core subunit of the integrator transcriptional complex enriched at UV damage sites, and DDX24 was found to interact with G4 quadruplex DNA [89,90,91]. DNA G4-quadruplex structures may underpin DDX helicase transcription initiation mechanisms and/or stabilize R-loop structures [92,93]. In fact, this highlights the notion that there may be non-overlapping steps to regulating these species. The large number of DDX helicases important for regulating DNA:RNA hybrids and/or R-loops may be related to the complex nature of such nucleic acid species (Figure 2). DDX helicases also functionally converge in ribosome biogenesis. Furthermore, while our review did not focus on the connection between DDX helicases and cell cycle regulation, this certainly affects genome stability and has been well reviewed elsewhere [94]. The related family of DEAH helicases also represent a fascinating intersection between RNA metabolism and genome stability, notably DHX9 [95,96]. Collectively, these helicases could serve as potential targets in combination with other therapeutic interventions of cancer.

## Figures and Tables

**Figure 2 genes-12-01471-f002:**
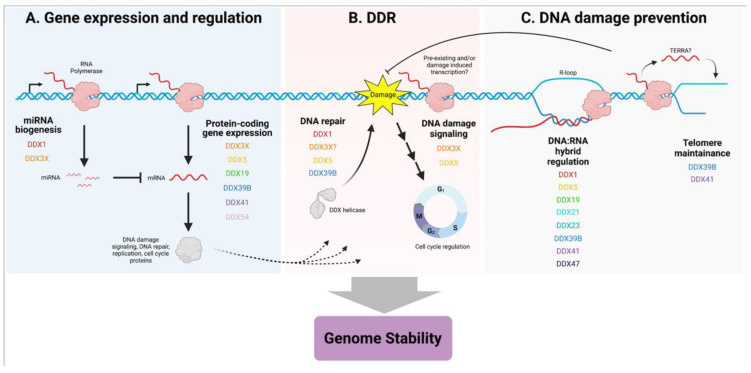
Summary of the themes regarding DDX RNA helicase involvement in genome stability. All DDX helicases mentioned are listed. (**A**) DDX RNA helicases promote the expression of genes important for genome stability (represented in the dotted lines) in addition to regulating their expression via microRNAs. (**B**) DDX RNA helicases are directly involved with the DDR either via DNA repair or the p53 DNA damage signaling axis, typically associated with the RNA metabolism aspects of these processes. (**C**) DDX helicases regulate complex nucleic acid species such as DNA:RNA hybrids and telomeric sequences, which are associated with DNA damage and genomic instability if left unprocessed. Created with BioRender.com accessed on 20 July 2021.

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
