# Peer review of "DEAD-Box RNA Helicases and Genome Stability"

_genes, 2021, doi:10.3390/genes12101471_

Round 1
Reviewer 1 Report
This review is timely and well written, it covers well the topic with some slight concerns exposed here.
Generally speaking, I feel that restricting the review to DEAD-box proteins was questionable, as some of highly related DEAH-box helicases have also been involved in similar mechanistical aspects related to genome integrity, which could have been at least mentioned, if not detailed, in the review. I am thinking for example about the well-established role of DHX9 on R-loop structures. Converserly, I did not find really interesting the paragraph on DDX54, which is quite speculative and for which very little is known.
Also, what strikes the reader is the strong overlap between the functions of the different DEAD box helicases, in particular their involvement in resolving R-loops. I would have appreciated a general discussion around these intriguing questions: how silencing experiments affecting single proteins can lead to the same phenotype (increased R-loops), while so many other helicases known to resolve them are still expressed ? What can explain this high level of functional redundancy ?
More minor points:
- I find Figure 2 a bit incomplete with respect to what is detailed in the main text. For example, DDX5 could be part of Fig.2A as an important regulator of CDKN1A (Nicol et al), and also be part of Fi.2C as a modulator of R-loops. Please explain better or complete the figure.
- Some typos remain in the text, for example in lines 196 '"deletion of Sub2") or 197 (hrp1/hpr1)
Author Response
We deeply appreciate your comments and feedback. We are glad you agree with the value of this review article.
We hope the following revisions adequately address your review (reviewer text bolded):
This review is timely and well written, it covers well the topic with some slight concerns exposed here.
Generally speaking, I feel that restricting the review to DEAD-box proteins was questionable, as some of highly related DEAH-box helicases have also been involved in similar mechanistical aspects related to genome integrity, which could have been at least mentioned, if not detailed, in the review. I am thinking for example about the well-established role of DHX9 on R-loop structures.
We agree there would be intellectual merit in including a review of DEAH-box helicases. The window between the invitation to write the review and the submission date was small, precluding a meaningful review of DEAH-box related literature. Thus, we narrowed our focus to DEAD-box RNA helicases. However, we have noted the importance of DEAH-box RNA helicase research in section 4 ‘Concluding remarks’ starting from line 453.
Converserly, I did not find really interesting the paragraph on DDX54, which is quite speculative and for which very little is known.
We have re-written the DDX54 discussion into section 3 ‘Implications for cancer’ starting at line 316.
Also, what strikes the reader is the strong overlap between the functions of the different DEAD box helicases, in particular their involvement in resolving R-loops. I would have appreciated a general discussion around these intriguing questions: how silencing experiments affecting single proteins can lead to the same phenotype (increased R-loops), while so many other helicases known to resolve them are still expressed ? What can explain this high level of functional redundancy ?
We have expanded on this fascinating effect in section 4 ‘Concluding remarks’ starting at line 446.
More minor points:
- I find Figure 2 a bit incomplete with respect to what is detailed in the main text. For example, DDX5 could be part of Fig.2A as an important regulator of CDKN1A (Nicol et al), and also be part of Fi.2C as a modulator of R-loops. Please explain better or complete the figure.
The absence of DDX5 is a typo. We have corrected Figure 2 and clarified the listing of DDX helicases in the caption.
- Some typos remain in the text, for example in lines 196 '"deletion of Sub2") or 197 (hrp1/hpr1)
We have resolved this typo among others seen in the track changes.
Reviewer 2 Report
Report on the review article “DEAD-box RNA helicases and genome stability”.
This review covers a very interesting and important topic that has been rarely, if ever, reviewed.
The authors have provided a thorough bibliographical compilation.
My main regret is that the authors opted for a “stamp collection” type of review i.e. listing sequentially all the information relevant to each helicase. I feel the review would have been greatly improved by structuring it according to the different processes these helicases are involved in, using the very nice Figure 2 as an illustration. I will not insist on this modification, unless the authors feel that can easily reorganise their review. However, I think the authors should make clear in their introduction that not all DEAD box helicases have been shown to dissociate short nucleic acid duplexes. In contrast, to my knowledge, they all display ATPase activity in vitro.
Minor points:
- Line 77: the following formulation seems incorrect to me: “… and only recently have had a DDX helicase described in this specialized process”.
- Line 92 : change to : “…miRNAs are used to rapidly modulate…”
- Line 95: change to “.. function of DDX1 in miRNA biogenesis…”
- Line 97: change to “of our understanding of the many roles…”
- Line 101 and the following lines: The sentence is difficult to understand. Some piece of information is missing. Do the authors refer to DDX3X-depleted conditions ?
- Lines 189-190: This sentence is very hard to understand. Do the authors refer to the relocation of the rDNA to the nucleolar periphery ?
- Line 350: delete “RX5902”.
Author Response
We are very glad the reviewer agrees with the importance of this work. We are very thankful for the time taken to review our manuscript and hope to incorporate their feedback adequately in the following revisions (reviewer text in black):
Report on the review article “DEAD-box RNA helicases and genome stability”.
This review covers a very interesting and important topic that has been rarely, if ever, reviewed.
The authors have provided a thorough bibliographical compilation.
My main regret is that the authors opted for a “stamp collection” type of review i.e. listing sequentially all the information relevant to each helicase. I feel the review would have been greatly improved by structuring it according to the different processes these helicases are involved in, using the very nice Figure 2 as an illustration. I will not insist on this modification, unless the authors feel that can easily reorganise their review.
We similarly debated both formats. We ultimately decided on the stamp collection format with a summary figure (Figure 2). In this way, researchers studying a specific (or related) helicase may quickly understand the research context of a helicase of interest, while still having access to a more holistic framework of their roles. While we agree there is also merit to the alternative format, we are also limited by the revision turnaround time.
However, I think the authors should make clear in their introduction that not all DEAD box helicases have been shown to dissociate short nucleic acid duplexes. In contrast, to my knowledge, they all display ATPase activity in vitro.
We have revised that passage accordingly starting at line 26.
Minor points:
Line 77: the following formulation seems incorrect to me: “… and only recently have had a DDX helicase described in this specialized process”.
We have removed that line for clarity starting at line 87.
Line 92 : change to : “…miRNAs are used to rapidly modulate…”
Revised at line 91.
Line 95: change to “.. function of DDX1 in miRNA biogenesis…”
Revised at line 103.
Line 97: change to “of our understanding of the many roles…”
Revised at line 94.
Line 101 and the following lines: The sentence is difficult to understand. Some piece of information is missing. Do the authors refer to DDX3X-depleted conditions ?
Revised starting at line 113.
Lines 189-190: This sentence is very hard to understand. Do the authors refer to the relocation of the rDNA to the nucleolar periphery ?
We have revised this passage for clarity starting at line 204.
Line 350: delete “RX5902”.
Revised accordingly at line 412.